

# Age, growth, and natural mortality of yellowfin grouper (*Mycteroperca venenosa*) from the southeastern United States

Michael L. Burton, Jennifer C. Potts and Daniel R. Carr

Beaufort Laboratory, Southeast Fisheries Science Center, National Marine Fisheries Service, NOAA, Beaufort, NC, USA

## ABSTRACT

Ages of yellowfin grouper ($n = 306$) from the southeastern United States coast from 1979–2014 were determined using sectioned sagittal otoliths. Opaque zones were annular, forming January–June (peaking in February–March). Yellowfin grouper ranged in age from 3 to 31 years; the largest fish measured 1,000 mm fork length (FL). Body size relationships for yellowfin grouper were: $W = 1.22 \times 10^{-5} \, \text{FL}^{3.03}$ ($n = 229$, $r^2 = 0.92$); $\text{TL} = 1.06 \, \text{FL} - 14.53$ ($n = 60$, $r^2 = 0.99$); and $\text{FL} = 0.93 \, \text{TL} + 18.63$ ($n = 60$, $r^2 = 0.99$), where $W$ = whole weight in grams, FL in mm, and TL = total length in mm. The von Bertalanffy growth equation was: $L_t = 958 \, (1 - e^{-0.11(t+2.94)})$ ($n = 306$). The point estimate of natural mortality for yellowfin grouper was $M = 0.14$, while age-specific estimates of $M$ ranged from 1.59 to 0.17 for ages 1–31.

## INTRODUCTION

The yellowfin grouper (*Mycteroperca venenosa* Linnaeus 1758), a moderate- to large-sized member of the family Serranidae, is widely distributed throughout the western Atlantic Ocean from North Carolina through the Florida Keys (referred hereafter as the southeastern US or SEUS), into the northern Gulf of Mexico, and in waters off Bermuda and throughout the Caribbean south to Brazil. Juveniles are often found in the shallow turtle grass (*Thalassia testudinum*) beds, while adults are typically found on subtropical rocky hardbottom and coral reef areas at depths up to 137 m (*Heemstra & Randall, 1993*). Yellowfin grouper feed mainly on fishes and squid (*Randall, 1967*) and are known to form spawning aggregations (*Kadison et al., 2010*; *Scharer et al., 2012*). *Cushion (2010)* studied growth and reproduction of specimens from the Bahamas.

Yellowfin grouper are of moderate importance to the SEUS reef fish fishery. While caught infrequently by anglers, their large size makes them a prized trophy species. Estimated total annual landings from headboats (vessels carrying at least seven anglers engaged in recreational fishing) sampled by the Southeast Region Headboat Survey (SRHS), conducted by the National Marine Fisheries Service (NMFS), averaged 304 kg from 1986 to 2013 (K Brennan, 2014, unpublished data). Annual numbers of fish landed by

Corresponding author
Michael L. Burton,
michael.burton@noaa.gov

anglers fishing from private recreational boats and charter boats, estimated by the NMFS Marine Recreational Information Program (MRIP, T Sminkey, 2014, unpublished data) averaged 649 fish from 1981 to 2013. Commercial fisheries of the SEUS on average annually harvested 2,163 kg of yellowfin grouper from 1981–2013 (N Baertlein, pers. comm., 2014), primarily from hook-and-line gear. Landings are widely distributed along the SEUS coast, from North Carolina through the Florida Keys, including the Dry Tortugas.

Yellowfin grouper are currently included in the South Atlantic Fishery Management Council's (SAFMC) Snapper-Grouper Fishery Management Plan (FMP). Since 1992 the species has been regulated by a 20 inch (508 mm) total length (TL) size limit in commercial and recreational fisheries; they have been included in a shallow-water grouper closed season from January 1 to April 30 of each year since 2012, and included in an aggregate three-grouper-per-person-per-day bag limit for recreational fishermen outside of the closed season since 2012 (five grouper bag limit during 1992–2011) (*SAFMC, 2015*). Commercial regulations include the 20 inch size limit and inclusion in the shallow-water grouper closure from January 1 to April 30 (*SAFMC, 2015*). Yellowfin grouper are not currently scheduled for a National Marine Fisheries Service (NMFS) stock assessment under the Southeast Data, Assessment and Review (SEDAR) program, likely due to low annual landings and management priorities.

Information about size-at-age and growth rates of reef fishes is important to fishery managers. The preferred method of aging reef fish is to use the sagittal otoliths, or ear stones (*Manooch, 1987*), a calcareous structure found inside the cranium. These sagittae may be read as whole structures but are usually sectioned into several thin sections and the sections looked at under a microscope to elucidate the age of the fish. Age is determined by counting alternating opaque and translucent bands deposited due to fluctuations in environmental conditions such as water temperature. Nonlinear regression relating the measured length of the fish to the estimated age leads to the generation of growth curves, which are one of the most important inputs into the stock assessment process used by NMFS to manage fisheries (K Siegfried, NMFS Beaufort Laboratory, pers. comm., 2015).

We studied yellowfin grouper because little is known of their life history in SEUS waters. The desirability of the species as a trophy fish for recreational angles due to its large size, the relative infrequency with which it is caught, and the potential for overexploitation by overfishing spawning aggregations all make it imperative to study the basic biology needed for proper fishery management. Herein, we describe age and growth parameters and natural mortality, which are important input variables for agency-led stock assessment efforts. This study provides the first published information on life history parameters for yellowfin grouper from SEUS waters.

## MATERIALS AND METHODS

### Age determination and timing of opaque zone formation

Yellowfin grouper ($n = 308$) were opportunistically sampled by NMFS and state agencies' port agents sampling the recreational headboat and commercial fisheries in the SEUS from 1979 to 2014. All specimens used in this study were killed as part of legal fishing operations

and were already dead when sampled by the port agents, thus all research was conducted in accordance with the Animal Welfare Act (AWA) and with the US Government Principles for the Utilization and Care of Vertebrate Animals Used in Testing, Research, and Training (USGP) OSTP CFR, May 20, 1985, Vol. 50, No. 97. All specimens were captured by either conventional vertical hook and line gear or longline gear. Fork lengths (FL) and TL of specimens were recorded in millimeters (mm). Whole weight (W) in grams (g) was recorded for fish landed in the headboat fishery, but information about sex was not routinely recorded due to time constraints. Fish landed commercially were eviscerated at sea, thus whole weights and information about sex were unavailable. Sagittal otoliths were removed during dockside sampling and stored dry in coin envelopes. Otoliths were sectioned in the transverse dorso-ventral plane on a low-speed saw, following the methods of *Potts & Manooch (1995)*. Three serial 0.5 mm sections were taken near the otolith core. Sections were mounted on microscope slides with thermal cement and covered with mounting medium before analysis. The sections were viewed under a dissecting microscope at 12.5× using transmitted light. Each sample was assigned an opaque zone, or ring, count by an experienced reader (MLB) (*Burton, 2001*; *Burton, 2002*; *Burton, Potts & Carr, 2012*). Sections were read with no knowledge of date of capture or fish size. To ensure consistency between readers in the interpretation of growth structures, a second reader (JCP) read a subset ($n = 102$) of slides, then we calculated between-reader indices of average percent error (APE) following the methodology of *Beamish & Fournier (1981)*. When annuli counts differed between paired readings, the initial reading was used.

Increment periodicity was assessed using edge analysis. The edge type of the otolith was noted: 1 = opaque zone forming on the edge of the otolith section; 2 = narrow translucent zone on the edge, generally <30% of the width of the previous translucent zone; 3 = moderate translucent zone on the edge, generally 30%–60% of the width of the previous translucent zone; 4 = wide translucent zone on the edge, generally >60% of the width of the previous translucent zone (*Harris et al., 2007*). Based upon edge frequency analysis, all samples were assigned a calendar age, obtained by increasing the opaque zone count by one if the fish was caught before that year's increment was formed and had an edge which was a moderate to wide translucent zone (types 3 and 4). Fish caught during the time of year of opaque zone formation with an edge type of 1 or 2 were assigned a calendar age equal to opaque zone count. All fish caught after opaque zone formation would have a calendar age equivalent to the opaque zone count.

## Growth

*von Bertalanffy (1938)* growth parameters were derived using PROC NLIN, a non-linear regression procedure using least squares estimation and the Marquardt iterative algorithm option, in SAS statistical software (vers. 9.3; *SAS Institute, Inc., 1987*). If the residuals of the un-weighted model appeared skewed at the tails of the sample distribution, we inverse-weighted the model by $1/n$ of each calendar age. Given studies showing that peak spawning of yellowfin grouper occurs March–April in the Bahamas and Caribbean (*Smith, 1961*; *Thompson & Munro, 1978*; *Cushion, 2010*; *Scharer et al., 2012*), we defined the

birthdate for all yellowfin grouper as April 1. To account for growth of the fish throughout the year before or after its "birthday," the calendar age of the fish was adjusted for the time of year caught ($\text{Mo}_c$), thus creating a fractional or monthly biological age ($\text{Age}_f$) from the calendar age ($\text{Age}_c$) based on the April 1 birthdate ($\text{Mo}_b$):

$$\text{Age}_f = \text{Age}_c + ((\text{Mo}_c - \text{Mo}_b)/12).$$

## Body-size relationships

For weight–length relationships, we regressed $W$ on TL ($n = 229$) and FL ($n = 59$), examining both a non-linear fit by using nonlinear least squares estimation (*SAS Institute, Inc., 1987*) and a linearized fit of the log-transformed data, examining the residuals to determine which regression was appropriate. For length–length relationships, we regressed TL on FL and FL on TL ($n = 60$) using linear regression.

## Natural mortality

We estimated the instantaneous rate of natural mortality ($M$) using two methods:
    (1) *Hewitt & Hoenig*'s (*2005*) longevity mortality relationship:

$$M = 4.22/t_{\max}$$

where $t_{\max}$ is the maximum age of the fish in the sample, and
    (2) *Charnov, Gislason & Pope*'s (*2013*) method using life history parameters:

$$M = (L/L_\infty)^{-1.5} \times K$$

where $L_\infty$ and $K$ are the von Bertalanffy growth equation parameters, when $t_0$ is assumed to be 0, and $L$ is fish length at age. The *Hewitt & Hoenig (2005)* method uses longevity to generate a single point estimate. The Charnov method, which incorporates life history information via estimated growth parameters, is based upon evidence suggesting that $M$ decreases as a power function of body size. This method generates age-specific rates of $M$ and is currently in use in SEDAR stock assessments (E Williams, NMFS Beaufort Laboratory, pers. comm., 2013).

# RESULTS

## Age determination and timing of opaque zone formation

A total of 308 sagittal otoliths of yellowfin grouper were sectioned. The distribution by area and fishery sector of samples used in the age analysis is shown in Table 1. The majority of samples came from the Carolinas ($n = 277$) with most ($n = 259$) from commercial fisheries (Table 1). The remaining samples ($n = 31$) came from Florida. We assigned an opaque zone count to 306 (99%) yellowfin grouper sections. Two specimens were excluded because sections were illegible.

We assigned an edge type to all readable samples for our analysis of increment periodicity. Yellowfin grouper deposited opaque zones on the otolith marginal edge January through June (Fig. 1), with peak formation in February and March. A transition

**Table 1 Geographic and fishery sector distribution of yellowfin grouper aging study.** Number of samples of sagittal otoliths that were used for age and growth study of yellowfin grouper (*Mycteroperca venenosa*) collected from 1979 to 2014 from fisheries landings along the coast of the southeastern United States. Samples were collected in the following states: North Carolina (NC), South Carolina (SC), and Florida (FL).

| State | Commercial | Recreational |
|---|---|---|
| NC | 134 | 3 |
| SC | 125 | 15 |
| FL | 3 | 28 |

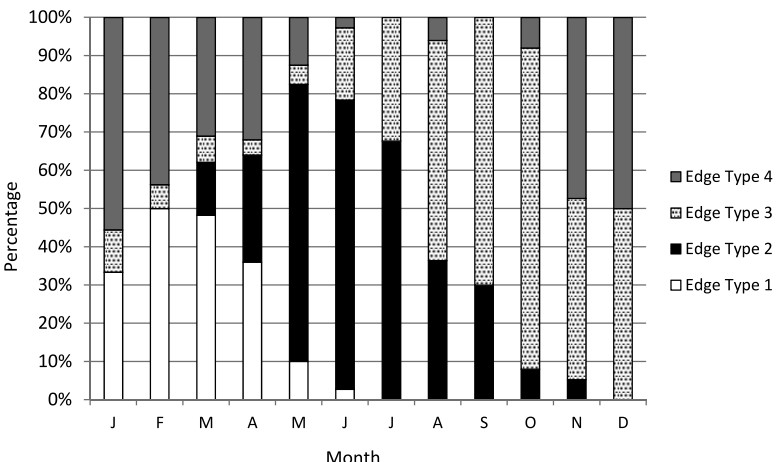

**Figure 1 Otolith edge analysis to determine timing of opaque zone formation.** Monthly percentage of edge types on yellowfin grouper Mycteroperca venenosa otoliths ($n = 306$). Edge codes: 1 = opaque zone on edge; 2 = small translucent zone, <30% of previous increment; 3 = moderate translucent, 30%–60% of previous increment; 4 = wide translucent, >60% of previous increment.

to a narrow translucent edge occurred beginning in May. Yellowfin grouper otoliths were without an opaque zone on the edge from July through December. The widest translucent edges occurred November and December, prior to opaque zone formation in January. We concluded that opaque zones on yellowfin grouper otoliths were annuli. Calendar ages based on edge analysis were assigned as follows: for fish caught January through June and having edge types of 3 and 4, the annuli count was increased by one; for fish caught in that same time period with edge types 1 and 2, as well as for fish caught from July to December, the calendar age was equivalent to the annuli count.

Yellowfin grouper annuli were relatively easy to interpret (Fig. 2). Agreement was good between readers for otolith sections from yellowfin grouper. Average percent error, or APE, was 3.32% ($n = 102$), which is less than *Campana*'s (*2001*) threshold level of acceptability of 5% for species of moderate longevity and reading complexity. Direct agreement between readings was moderate (44%), and agreement for ±1 year was 83%. The largest discrepancy between readings was a difference of three, for a 16 year old fish.

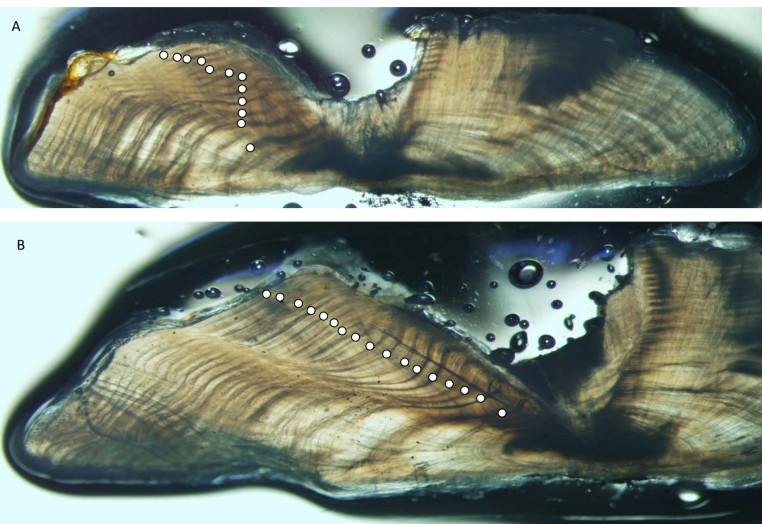

**Figure 2 Photographs of sections from two yellowfin grouper.** Sections from otoliths of yellowfin grouper *Mycteroperca venenosa*: (A) 720 mm FL, age 12 yrs: (B) 915 mm FL, age 17 yrs. Age was determined by counting opaque increments along the dorsal axis and sulcus using transmitted light at 12.5X magnification.

## Growth

Yellowfin grouper in this study ranged from 405 to 1,000 mm FL and ages 3–31, although only 12 fish were older than age-19 (Table 2). The resulting von Bertalanffy growth equation for the unweighted, freely-estimated model was:

$$L_t = 958(1 - e^{-0.11(t+2.94)}) \quad (n = 306; \text{Fig. 3}).$$

Predicted size-at-age from this model run agreed well with mean observed size-at-age (Fig. 4). Fish less than age-3 were unavailable to us, no doubt because hook-and-line gear generally select for larger fish and the minimum size limit regulations excluded the smaller fish from the landings. Consequently, the model was unable to depict initial growth of young fish, thus explaining the moderately negative value of $t_0$. Therefore, we re-estimated the growth models using a fixed value of $t_0 = -0.5$ (*Burton, Potts & Carr, 2012*), which has the effect of pulling the growth curve down to simulate smaller fish length at age for the youngest ages. The value of $-0.5$ was selected to approximate growth of age-0 fish, which is an annual age that encompasses twelve months. This modelling procedure is currently in use in SEDAR stock assessments (*SEDAR, 2013*) for species that either have a regulatory minimum size limit or exhibit size selectivity toward fishing gear. The model was inverse weighted to adjust for the lower limb of the curve being pulled down. The residuals for the largest, oldest fish were more evenly distributed in this model. The resulting growth model is:

$$L_t = 929(1 - e^{-0.156(t+0.5)}), \quad (n = 306; \text{Fig. 3}).$$

**Table 2 Age-specific observed and predicted size and natural mortality for yellowfin grouper from the SEUS.** Observed and predicted mean fork length (FL) from the freely estimated growth model, measured in millimeters, and natural mortality at age (M, *Charnov, Gislason & Pope, 2013*) data for yellowfin grouper (*Mycteroperca venenosa*) collected from 1979 to 2014 along the coast of the southeastern United States. Standard errors of the means (SE) are provided in parentheses.

| Age | n | Mean FL (±SE) | FL range | Predicted FL | M |
|---|---|---|---|---|---|
| 1 | | | | 308 | 1.59 |
| 2 | | | | 380 | 0.83 |
| 3 | 4 | 428 (11) | 405–453 | 445 | 0.57 |
| 4 | 9 | 533 (20) | 413–600 | 502 | 0.44 |
| 5 | 20 | 583 (13) | 457–703 | 553 | 0.36 |
| 6 | 30 | 618 (11) | 500–820 | 599 | 0.31 |
| 7 | 22 | 649 (9) | 537–710 | 639 | 0.28 |
| 8 | 23 | 687 (11) | 597–810 | 675 | 0.25 |
| 9 | 21 | 717 (13) | 617–820 | 707 | 0.24 |
| 10 | 22 | 736 (7) | 674–788 | 735 | 0.22 |
| 11 | 20 | 753 (11) | 635–830 | 760 | 0.21 |
| 12 | 30 | 782 (10) | 644–850 | 783 | 0.21 |
| 13 | 24 | 793 (10) | 692–872 | 803 | 0.20 |
| 14 | 14 | 828 (13) | 742–910 | 821 | 0.19 |
| 15 | 14 | 821 (13) | 705–880 | 836 | 0.19 |
| 16 | 19 | 847 (13) | 754–960 | 851 | 0.19 |
| 17 | 7 | 874 (11) | 840–915 | 863 | 0.18 |
| 18 | 8 | 871 (17) | 805–928 | 874 | 0.18 |
| 19 | 7 | 884 (18) | 823–940 | 884 | 0.18 |
| 20 | 3 | 882 (27) | 850–935 | 893 | 0.18 |
| 21 | 3 | 880 (48) | 850–935 | 901 | 0.18 |
| 22 | 2 | 922 (37) | 885–958 | 908 | 0.18 |
| 23 | | | | 914 | 0.17 |
| 24 | 1 | 998 | | 920 | 0.17 |
| 25 | 1 | 870 | | 924 | 0.17 |
| 26 | | | | 929 | 0.17 |
| 27 | | | | 933 | 0.17 |
| 28 | 1 | 1,000 | | 936 | 0.17 |
| 29 | | | | 939 | 0.17 |
| 30 | | | | 942 | 0.17 |
| 31 | 1 | 900 | | 944 | 0.17 |

In 1992 a 20-inch (508 mm) minimum size limit enacted for yellowfin grouper excluded smaller fish from the landings and thus our samples. We re-ran the growth model using the method of *McGarvey & Fowler (2002)*, which adjusts for the bias imposed by minimum size limits by assuming zero probability of capture below the minimum size limit. The resulting von Bertalanffy growth equation was:

$$L_t = 966(1 - e^{-0.13(t+2.48)}) \quad (n = 305; \text{Fig. 3}).$$

## Body-size relationships

Statistical analyses revealed a multiplicative error term (variance increasing with size) in the residuals of the $W$–TL and $W$–FL relationships for yellowfin grouper, indicating a linearized ln-transform fit of the data was appropriate. The relationships are described by the following regressions:

$$\ln(W) = 3.026 \times \ln(\text{TL}) - 11.345 \quad (n = 229, r^2 = 0.92)$$
$$\ln(W) = 2.915 \times \ln(\text{FL}) - 10.453 \quad (n = 59, r^2 = 0.98).$$

These equations were transformed back to the form $W = a(\text{L})^b$ after adjusting the intercept for log-transformation bias with the addition of one-half of the mean square error (1/2 MSE) (*Beauchamp & Olson, 1973*), resulting in the relationships

$$W = 1.22 \times 10^{-5}\text{TL}^{3.03} \quad \text{and}$$
$$W = 2.89 \times 10^{-5}\text{FL}^{2.91}.$$

The relationships between TL and FL are described by the equations

$$\text{TL} = 1.06 \times \text{FL} - 14.53 \quad (n = 60; r^2 = 0.99) \quad \text{and}$$
$$\text{FL} = 0.93 \times \text{TL} + 18.63 \quad (n = 60; r^2 = 0.99).$$

## Natural mortality

Natural mortality ($M$) was estimated at 0.14 using the method of *Hewitt & Hoenig (2005)*, integrating all ages into a single point estimate and using the maximum age from our study of age 31. Because *Charnov, Gislason & Pope*'s (*2013*) age-specific calculation of $M$ assumed a von Bertalanffy growth function with $t_0 = 0$, we re-estimated $K$ and $L_\infty$ with the constraint $t_0 = 0$, and inverse weighting the model. The resulting parameters were $L_\infty = 921$ mm TL and $K = 0.17$. Age-specific estimates of $M$ using *Charnov, Gislason & Pope (2013)* are presented in Table 2. We used the midpoint of each age (e.g., 0.5, 1.5, 2.5, etc.) to calculate age-specific $M$, because the *Charnov, Gislason & Pope (2013)* method cannot mathematically calculate $M$ for absolute age-0. Also, for stock assessment purposes where the integer age is used to describe the entire year of the fish's life, the mid-point gives the median value of $M$ for that age.

## DISCUSSION

Otolith edge analysis demonstrated that yellowfin grouper deposited one annulus per year between January and June, with peak annulus formation between February and March. This is similar to timing of annulus formation for other groupers in the SEUS, which tend to form annuli in winter (*Moe, 1969*; *Manooch & Haimovici, 1978*; *Burton, Potts & Carr, 2012*). *Crabtree & Bullock (1998)* found that the congeneric black grouper (*Mycteroperca bonaci*), formed annuli from April to June in Florida waters.

Yellowfin grouper grew moderately fast, attaining an average observed size of 428 mm FL by age-3, 583 mm by age-5, 736 mm by age-10, and 821 mm by age-15. Subsequently,

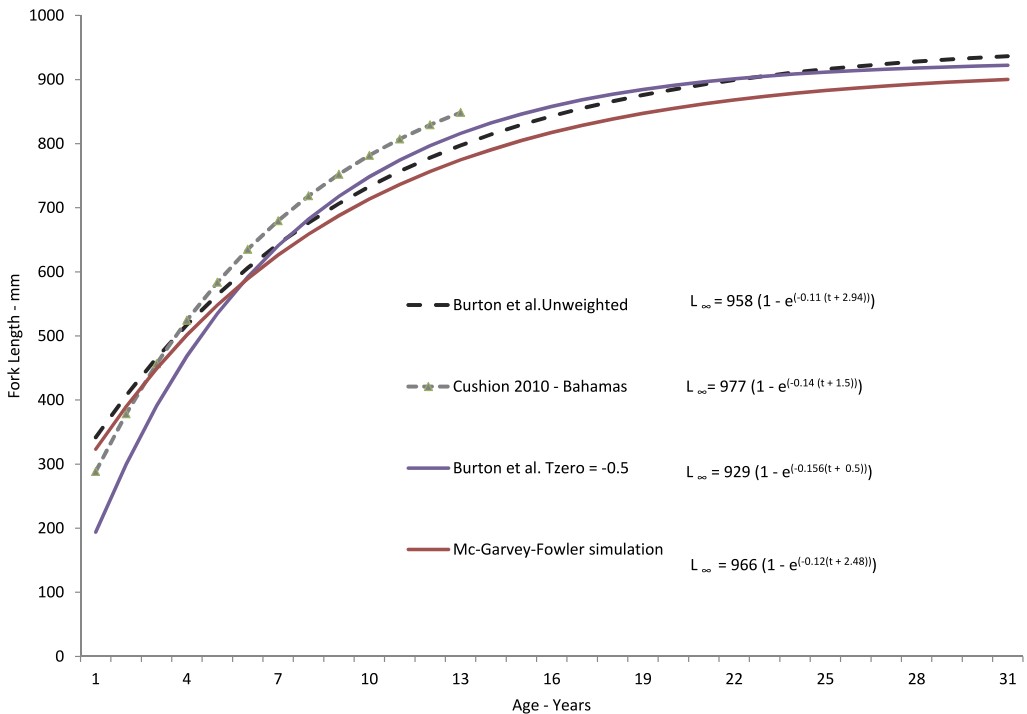

**Figure 3  Comparison of theoretical growth of yellowfin grouper from various studies.** Comparison of SEUS yellowfin grouper von Bertalanffy growth curves for freely estimated (unweighted), $t_0$-restrained at −0.5, and size limit-corrected model runs (*McGarvey & Fowler, 2002*). Growth model from the Bahamas population (*Cushion, 2010*) is presented for comparison.

growth slowed to an average of 14 mm per year (Table 2). Observed size at age for yellowfin grouper from the SEUS compared favorably through age-13 with that of the Bahamian population studied by *Cushion (2010)* (Fig. 3) but our study found a much greater maximum age than that found by *Cushion (2010)*, 31 yrs as compared with 13 yrs for the Bahamian population. While both studies were comprised of fishery-dependent samples, we feel our study was more representative of the population we sampled due to broader geographic coverage and larger sample size. All of the samples from the Bahamas study came from a single fish-market in New Providence Island.

Our predicted growth curve of yellowfin grouper using the parameters from the freely estimated, unweighted growth model fit the observed data well (Fig. 4). The von Bertalanffy parameter *K*, or the Brody growth coefficient, which estimates the rate of attainment of maximum size, was lower in our study, 0.11, compared to 0.14 for Bahamian yellowfin grouper (*Cushion, 2010*). Conversely, maximum predicted length was slightly larger for Bahamian fish (977 mm) versus our study (958 mm), which is interesting considering maximum observed ages were age-13 and age-31, respectively. Curiously, the usual expectation is that tropical populations of fish grow faster and reach smaller maximum sizes and ages than subtropical or temperate populations (*Longhurst & Pauly, 1987*; *Berrigan & Charnov, 1994*). *Manickchand-Heileman & Phillip (2000)* pointed out that yellowmouth grouper (*Mycteroperca interstitialis*) populations from Trinidad and Tobago had larger maximum sizes, lower growth rates, and greater observed age when

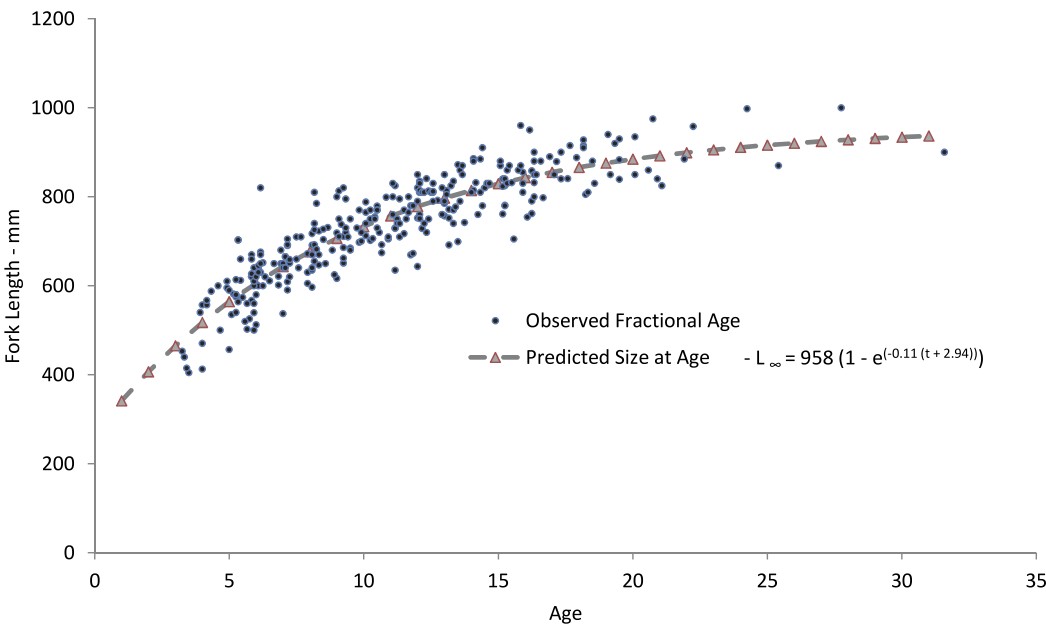

**Figure 4** **Comparison of observed and predicted size-at-age for yellowfin grouper from the current study.** Comparison of mean observed size at age (yrs) and sizes predicted by the von Bertalanffy freely-estimated growth model for yellowfin grouper *Mycteroperca venenosa* from the southeastern US.

compared with subtropical/temperate populations. They suspected that this was likely because of a shorter period of exploitation. Relatively high rates of fishing pressure in the tropics may tend to truncate the population age structure of yellowfin grouper in the Bahamas. Alternately, we examined almost four times as many specimens as *Cushion (2010)*. Perhaps with a greater sample size there is a greater chance of encountering fish of greater ages; *Hoenig (1983)* recognized this possibility and incorporated sample size into his estimator for mortality rates.

We constrained $t_0$ to $-0.5$ in our modified growth curve for yellowfin grouper, which had the effect of dampening the growth curve in the earliest years. However, by age-6 the modified curve and the freely estimated curve were nearly identical, differing only in the earlier ages. Adjustment of the curve for size limits using the method of *McGarvey & Fowler (2002)* resulted in very similar growth parameters as generated by the freely estimated model. While size at age-1 was slightly lower, the method did not pull the curve down in the earlier ages as much as the previous fixed $t_0$ model run did, and the von Bertalanffy growth parameters were very similar (Fig. 3). This result is likely explained by the fact that there were very few fish in the age classes most affected by the size limit prior to the implementation of the minimum size limit. Thus, the *McGarvey & Fowler (2002)* method could not fully determine and correct for the potential non-normal distribution of size-at-age.

Natural mortality ($M$) of wild fish populations is difficult to measure but is an important input into stock assessments. A point estimate of $M$ (*Hewitt & Hoenig, 2005*) for the entire life span of a fish seems unreasonable, because as fish grow they become less vulnerable to predation. We thought that our point estimate of $M$ was reasonable for fully recruited ages in our study but was an insufficient estimate of $M$ for all ages. The age-varying

$M$ calculated using *Charnov, Gislason & Pope (2013)* seems a more appropriate estimator for the younger ages. The initial Charnov estimates of $M$ starting with the fully recruited age-6 are approximately $2.6\times$ the Hewitt and Hoenig estimate, reflecting higher natural mortality at younger ages. The age-specific estimates of $M$ for the older ages stabilized near the *Hewitt & Hoenig (2005)* estimate of $M$ (Table 2). When considering the cumulative estimate of survivorship to the fully recruited ages, the *Hewitt & Hoenig (2005)* method estimated 2.9% survivorship, while the Charnov estimate was 0.6%. Very few of the fish in our samples were older than age-18 (19 of 306), and even fewer were older than age-22 ($n = 4$). Though sample size in this study was limited, the age-frequency distribution suggests that the chance of survivorship to the oldest age may truly be as low as 1%. There is no evidence that hook and line gear is dome-selective for this species or its congeners: thus our study had the potential to collect the largest and oldest fish in the population. These observations give weight to the argument to use Charnov's estimate of $M$ at age.

One limitation of this study was the lack of fish smaller than about 428 mm FL (or about age-3), because of the fishery-dependent nature of our samples, the selectivity of fishing gear, and the minimum size limits in place for yellowfin grouper. Lack of smaller fish is common in studies dominated by fishery-dependent samples and can lead to problems in estimating the growth curve for the youngest ages. Inclusion of fishery-independent samples usually corrects this problem, as fishery-independent gear such as traps will catch smaller fish. However, only two yellowfin grouper have been caught by the two major fishery-independent surveys operated by natural resource agencies in the southeast (Southeast Fishery Independent Survey, administered by the NOAA Fisheries/SEFSC/Beaufort Laboratory, 2010-present; Marine Monitoring, Assessment and Prediction Program annual survey, administered by South Carolina Department of Natural Resources, 1972-present). Younger fish were unavailable to us to help define the trajectory of the growth curve at the earliest ages, and this section of the growth curve should be interpreted with caution. We accounted for this limitation by re-estimating our growth parameters using a fixed value for $t_0$ of $-0.5$.

Another limitation of our study is the long period of time over which samples were collected ($>30$ yrs). Population parameters can vary inter-annually for various reasons (e.g., variable recruitment, environmental reasons), and it is certain that parameter estimates based on samples collected over 30 yrs would have increased variability when compared to estimates generated from samples collected over a much shorter time period. Unfortunately, samples from infrequently-caught species such as yellowfin grouper will likely never be obtained in quantities large enough to allow us to eliminate this source of error in the parameter estimates.

## CONCLUSIONS

This study is the first published study of yellowfin grouper life history in the SEUS. We have shown that otolith sections of yellowfin grouper contain annuli that are relatively easy to enumerate and that otolith sections are therefore likely reliable structures for aging. Growth rings on yellowfin grouper sagittae are assumed to be deposited once a year in

spring and growth is generally fast for the first seven years and then slows considerably, as evidenced by the low value of $K$, the von Bertalanffy growth coefficient. Our estimates of $M$ are reasonable for a fish with a moderately long life span and longevity to age 31. We believe the results of this study accurately describe the fished population of yellowfin grouper in the SEUS. The overall landings of this species in the commercial and recreational fisheries of the SEUS make it an unlikely candidate for a stock assessment through the NMFS SEDAR process because of the prioritization of more commonly landed species. A possible use of these data would be their application to studies of the population dynamics of US Caribbean stocks (US Virgin Islands and Puerto Rico). The US Caribbean is typically a data-poor region, and studies from the SEUS could be used as proxies in analyses for the region. However, analyses should be undertaken to determine appropriateness of such a procedure (i.e., similar life history traits between populations). Application of the growth curve from *Cushion*'s (*2010*) Bahamian population to populations from the wider Caribbean might not be warranted based on the low maximum age in her study vs. what the current study found. Precaution should always be taken when extrapolating beyond the scope of current data.

## ACKNOWLEDGEMENTS

We gratefully acknowledge the many headboat and commercial port samplers over the years whose efforts made this study possible. J Smith, R Cheshire, T Kellison and two anonymous reviewers provided valuable reviews that greatly improved the manuscript.

### Funding

This work was approved and funded by the National Marine Fisheries Service, Southeast Fisheries Science Center, Miami, FL and Beaufort Laboratory. There were no external grants. The manuscript was reviewed in-house by both the Laboratory and the Center prior to journal submission. The funders had no role in study design, data collection and analysis, decision to publish, or preparation of the manuscript.

### Grant Disclosures

The following grant information was disclosed by the authors:
National Marine Fisheries Service, Southeast Fisheries Science Center.
Beaufort Laboratory.

### Competing Interests

The authors declare there are no competing interests.

### Author Contributions

- Michael L. Burton conceived and designed the experiments, performed the experiments, analyzed the data, wrote the paper, prepared figures and/or tables, reviewed drafts of the paper.

- Jennifer C. Potts conceived and designed the experiments, performed the experiments, analyzed the data, contributed reagents/materials/analysis tools, wrote the paper, prepared figures and/or tables, reviewed drafts of the paper.
- Daniel R. Carr performed the experiments, processed samples, conducted literature review.

### Animal Ethics

The following information was supplied relating to ethical approvals (i.e., approving body and any reference numbers):

The following information was supplied relating to ethical approvals (i.e., approving body and any reference numbers):

All research was conducted in accordance with the Animal Welfare Act (AWA) and with the US Government Principles for the Utilization and Care of Vertebrate Animals Used in Testing, Research, and Training (USGP) OSTP CFR May 20, 1985, Vol. 50, No. 97.

The study was conducted on cold-blooded vertebrates (fish) which were already dead when collected and processed by the samplers for this study.

### Supplemental Information

Supplemental information for this article can be found online at http://dx.doi.org/10.7717/peerj.1099#supplemental-information.

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
