# Peer review of "Age, growth, and natural mortality of yellowfin grouper (Mycteroperca venenosa) from the southeastern United States"

_PeerJ, doi:10.7717/peerj.1099_

## Round 0.1 · original submission · Minor Revisions

· Academic Editor

Minor Revisions

Please, considered all the suggestions of the reviewers in your revised manuscript.

Reviewer 1 ·

Basic reporting

This study reports basic information on the age, growth and mortality of yellowfin grouper. Overall the paper is well written and provides useful life history parameters on a species from which few biological samples are available.

Experimental design

Results are based on an observational study. The methods are described in detail, are well established and appropriate.

Validity of the findings

The findings are statistically sound and the conclusions are presented appropriately.

Additional comments

This study reports basic information on the age, growth and mortality of yellowfin grouper. Overall the paper is well written and provides useful life history parameters on a species from which few biological samples are available. I recommend publication after some minor revision. Specific comments are listed by section below.

Introduction

Line 46: Delete “waters of”.
Line 58: Insert “on average” after SEUS.

Methods

Line 78: Edge type analysis methods used in this study are not validation of annulus formation (see Beamish and McFarlane 1983; Campana 2001). Sub-heading should be changed to “timing of opaque zone formation”.
Was sex recorded for any of the recreational samples?
Line 98: Change citation to Beamish and Fournier 1981.
Line 105: Delete “chronological or”. Calendar age is the most logical description.
Line 110: Change “chronological” to calendar
Lines 126-128: For FL and TL comparisons, why would you expect a non-linear fit?

Results

Line 144: Change “validation of annuli” to “timing of opaque zone formation”.
Line 155: Change “concluded” to “assumed”.
Lines 156 and 159: Change “chronological” to “calendar”.
Line 163: insert after 5% “for species of moderate longevity and reading complexity”.
Line 172: Why was a t0 of -0.5 selected instead of 0?

Discussion

Line 228: Insert “Conversely” at beginning of second sentence.
Line 280: After “are” insert “assumed to be”.

References

Line 356: Delete “DAT”.

Reviewer 2 ·

Basic reporting

Overall, this manuscript provides the first estimates of age, growth, and natural mortality of yellowfin grouper from the southeast US. This information will be extremely useful as fishery management moves towards data-limited stock assessments models in the interim until more data is collected. The authors do an excellent job of discussing potential issues with the current study, for example, aggregating aging samples over 3 decades. One suggestion is to portray the landings discussed in the introduction (lines 52-61) as a figure so the reader can interpret inter-annual variations in catches. This information could be easily obtained from NOAA commercial statistics - https://www.st.nmfs.noaa.gov/st1/commercial/landings/annual_landings.html.

Experimental design

Analyses within this manuscript are statistically sound and described in detail. My only concern here is that ages are determined by a single reader. This reader examines otoliths two separate times (2 months apart). The authors discuss this process as independent, which I don't think is accurate. Generally, more than 1 reader is used to read otoliths to account for reader bias.

Validity of the findings

The results of the study are based on scientifically-sound data. The authors do an excellent job of highlighting areas of uncertainty within their results and do not make any bold claims not supported by their data.

Additional comments

Although this manuscript is extremely well written, there are some areas for improvement. I have made suggestions and edits below to address formatting inconsistencies in the manuscript and reference section.

Minor comments

Lines 34-35: Change “Point estimates” to “The point estimate of…” and “were” to “was” since you are referring to a single estimate here.

Line 36: Replace 0.12 with 0.17 since 0.17 is shown in Table 2.

Lines 52-61: This information could be conveyed in a figure, which would enable the reader to ascertain inter-annual trends in landings by the recreational and commercial fleets. Which commercial gears capture yellowfin?

Lines 55, 59, 140, etc: Be consistent with first name initials and periods.

Line 59: Add space between “-2013”.

Line 64: Define “TL” here and use abbreviation on Line 86 since this is the first mention.

Lines 58, 59, 65, 67, 69 and throughout manuscript: Be consistent with spacing between numbers and hyphens.

Line 115 and throughout manuscript: Italicize “n”.

Lines 131, 198, 255, 256: Should “&” be spelled out?

Line 147: Suggested edit: “… (n = 277) with most (n = 250) from commercial fisheries” and italicize all ‘n’s.

Lines 152-153: It looks like the transition to a narrow translucent edge occurred in May and June also, since the majority of edges were narrow translucent during these months.

Line 154: Delete ‘amount of’.

Line 161: I’m not sure independent is accurate here since the same person read the otoliths during both time periods. It was my understanding that more than 1 reader is often used so that reader bias can be quantified?

Lines 163-164: What was the largest discrepancy between readings?

Line 169: Replace “select for generally larger fish” with “generally select for larger fish”.

Line 178: “excluded available smaller fish from our samples” reword, awkward as written.

Line 185: Replace “W - FL” with “W – FL” for consistency.

Line 187: Remove extra spacing in equation before “3.026”.

Lines 267: Do any surveys capture younger fish? Or have age-0s never been found?

References: Be consistent with either “-” or “–” between journal pages, bolding of volume, and period after journal name.

Lines 311-312, 327-328: Write out journal name for consistency.

Lines 314, 347, 368, 395: Remove comma in author names.

Line 342: Insert comma before Rome.

Lines 364-366: Delete “U.S. National Marine Fisheries Service” since you don’t use this in any other Fishery Bulletin reference?

Line 373: Remove period ”of.Marine” .

Table 1: A column and row showing totals would help the reader with interpreting this table.

Table 2: In “FL range” column, be consistent with either “-” or “–”.

Line 440: Italicize “n”.

Line 449: Add “Mycteroperca venenosa”.

Lines 449-451: Insert “(unweighted)” after “freely estimated” and “(McGarvey Fowler)” after “size limit-corrected model runs” since these are the labels used in Figure 4.

Figure 1: Labelling bars with months rather than numbers would help with interpretation.

Figure 3: It is difficult to tell the difference between the 2 gray lines; why not use black? Also, there appears to be a difference in font size between 4 equations shown on figure.

Figure 4: Which von B model was used here for prediction? Your line (dashed, with triangle) matches the Cushion model, but I don’t think this is the one you used?

·

Basic reporting

The manuscript is well written and concise. However, the introduction is a bit too brief, and only provides information on the fish and its fisheries, with no introduction to otoliths or any of the methods used. Given the wide subject matter and readership of PeerJ, some brief introduction on these might be appropriate.

Similarly, while the only other study on the growth of yellowfin grouper by Cushion (2010) was cited in the introduction, no details of those findings were outlined. These are highly relevant to the current study and a brief description should be included.

The body-size relationships were not mentioned in either the introduction or discussion and their purpose in the paper and usefulness to readers is questionable.

All figures and tables were appropriate and clear; although, fig. 4 is not referred to until the discussion and should be cited in the results. The labels on the different fitted lines in fig. 3 differ from those in the figure legend and text (e.g. 'Burton et al. Unweighted' vs 'freely estimated von Bertalanffy') - standardising this terminology would make it easier for the reader. Likewise the legend and labels of fig. 4 are a bit confusing - the description in the discussion is much clearer).

The paper currently lacks a conclusion section (as per submission guidelines) and one would help the reader to understand 'how the paper fits into the broader field of knowledge'. Likewise, although four growth equations are described and discussed in the paper, the authors provide no clear conclusion on which one is the most useful and should be applied by other researchers in future studies.

Experimental design

The research question is not very clearly outlined in paragraph four of the introduction. The paragraph suggests that the age and growth parameters are important variables for modelling and defining functional groups of reef fish - but the end of the preceding paragraph and final paragraph of the discussion indicates that due to very low landings the species is unlikely to be a candidate for stock assessment purposes in the region studied. The application of results to defining functional groups is never mentioned again in the paper. A more definitive statement of why the study took place and the meaningfulness of the research question in both the introduction and conclusions would greatly strengthen the paper (e.g. given their large body size, importance as trophy fish for recreational and charter boat fishers, potential for exploitation of spawning aggregations(?) and the difficulty in obtaining samples - this study analysed data and otoliths opportunistically collected over many years to provide age and growth parameters in temperate and subtropical waters for comparison with previous estimates derived from tropical populations - just one suggested approach). I'm not questioning the importance of the results, just how that importance is currently explained. At the end of the discussion, one might wonder why the results of Cushion (2010) that were collected from the tropics, aren't more appropriate and sufficient to inform management and stock assessments for Caribbean stocks? This seems to be the only relevance of the results provided.

The methods, results and ethical aspects were all clearly explained and well written. Although the dataset is small and collected over many years, this limitation and possible implications are adequately discussed.

Validity of the findings

The methods, data and results are all sound and robust. I would only suggest that geometric mean regression or standard major axis regression would be more appropriate than least squares estimation for the body-size relationships, given the likely measurement error in both variables. This reanalyses could be very easily completed, but is unlikely to change the results much.

Additional comments

Overall, it is a clear, easy to read paper, but a stronger introduction and conclusions clearly outlining the implications and relevance of the findings would make it more interesting and informative for readers.

---

## Round 0.2 · accepted · Accept

· Academic Editor

Accept

Authors have revised properly the manuscript.